# Roles of Ghrelin and Leptin in Body Mass Regulation under Food Restriction Based on the AMPK Pathway in the Red-Backed Vole, *Eothenomys miletus*, from Kunming and Dali Regions

**DOI:** 10.3390/ani12233333

**Published:** 2022-11-28

**Authors:** Yuting Liu, Ting Jia, Yue Ren, Zhengkun Wang, Wanlong Zhu

**Affiliations:** 1Key Laboratory of Ecological Adaptive Evolution and Conservation on Animals-Plants in Southwest Mountain Ecosystem of Yunnan Province Higher Institutes College, School of Life Sciences, Yunnan Normal University, Kunming 650500, China; 2Department of Pharmacy, Yunnan College of Business Management, Kunming 650106, China; 3College of Plant Protection, Shanxi Agricultural University, Taiyuan 030024, China; 4Engineering Research Center of Sustainable Development and Utilization of Biomass Energy, Ministry of Education, Kunming 650500, China; 5Key Laboratory of Yunnan Province for Biomass Energy and Environment Biotechnology, Kunming 650500, China

**Keywords:** *Eothenomys miletus*, ghrelin, leptin, food restriction, AMPK pathway

## Abstract

**Simple Summary:**

Animals can adapt to food shortages through phenotypic changes. In this study, food restriction reduced the body mass, RMR, serum leptin level, and POMC and CART gene expression in the red-backed vole, *Eothenomys miletus*; increased AMPK activity; and increased the concentration of ghrelin in the serum and stomach. The ghrelin-and-leptin-combined AMPK signaling pathway in the hypothalamus could play a role in the body mass regulation of *E. miletus* under food restriction. Moreover, regional differences in physiological indicators under food restriction may be related to the different temperatures or food resources in different regions.

**Abstract:**

The phenotype plasticity of animals’ physiological characteristics is an important survival strategy to cope with environmental changes, especially the change in climate factors. Small mammals that inhabit seasonally changing environments often face the stress of food shortage in winter. This study measured and compared the thermogenic characteristics and related physiological indicators in the adenosine-5′-monophosphate-activated protein kinase (AMPK) pathway in *Eothenomys miletus* between Kunming (KM, *n* = 18) and Dali (DL, *n* = 18) under food restriction and refeeding. The results showed that food restriction and the region have significant effects on body mass, the resting metabolic rate (RMR), hypothalamic neuropeptide gene expression, ghrelin levels in the stomach and serum, serum leptin level and the activity of AMPK, and malonyl CoA and carnitine palmitoyltransferase 1 (CPT-1) activity. Food restriction reduced the body mass, the gene expression of neuropeptide proopiomelanocortin (POMC), cocaine- and amphetamine-regulated transcription peptide (CART), and leptin level. However, the ghrelin concentration and AMPK activity increased. After refeeding, there was no difference in these physiological indexes between the food restriction and control groups. Moreover, the physiological indicators also showed regional differences, such as the body mass, POMC and CART gene expression, ghrelin concentration in the stomach and serum, and AMPK activity in DL changed more significantly. All these results showed that food restriction reduces energy metabolism in *E. miletus*. After refeeding, most of the relevant physiological indicators can return to the control level, indicating that *E. miletus* has strong phenotypic plasticity. Ghrelin, leptin, and the AMPK pathway play an important role in the energy metabolism of *E. miletus* under food restriction. Moreover, regional differences in physiological indicators under food restriction may be related to the different temperatures or food resources in different regions.

## 1. Introduction

Phenotypic plasticity usually refers to the ability of animals to exhibit different and adaptive characteristics in response to altered environmental conditions [1,2,3]. Such plastic changes can occur at the morphological, physiological, and behavioral levels, making animals better fit in a particular environment that they encounter. Although there have been many studies on phenotypic changes and their importance in adaptation and evolution has been well discussed, the underlying mechanisms remain unclear [4,5].

Animals can adapt to food shortages through phenotypic changes [6]. Many endocrine factors and hypothalamic neuropeptide have been implicated in regulating phenotypic changes associated with food shortage [7]. Previous studies have shown that under the condition of food shortage, the body mass, resting metabolic rate (RMR), and leptin decrease, while ghrelin and adenosine-5′-monophosphate-activated protein kinase (AMPK) activity increases; among them, the hypothalamic AMPK signaling pathway is the key to regulating animals’ feeding [8], which is mainly manifested in that food restriction increases AMPK activity in multiple regions of the hypothalamus, which decreases after refeeding [9]. For example, the research data of *Rattus norvegicus* showed that food shortage increases AMPK activity in the hypothalamus and stimulates feeding and body mass gain, while inhibiting AMPK activity in the hypothalamus after refeeding, leading to anorexia and body mass loss [10].

Appetite-stimulating hormones, such as ghrelin, adiponectin, and thyroid hormones, can activate the hypothalamic AMPK signaling pathway [9,11,12,13]. Among them, ghrelin is a stomach-derived hormone composed of 28 amino acids, which is secreted by X/A-like cells from the neck of the stomach bottom to the bottom of the oxygenic gland, and its mRNA is highly expressed in stomach tissue, with low-level expression in the intestine, pancreas, kidney, and placenta [14]. It binds to growth hormone secretagin receptor 1A (GHS-R1a), which can promote food intake and appetite, thereby regulating the body mass [15]. Food restriction increased AMPK activity by increasing ghrelin secretion, thereby activating the AMPK signaling pathway [9,13]. Anorexic hormones, such as leptin, insulin, and glucagon-like peptide-1 (GLP-1), can inhibit the hypothalamic AMPK signaling pathway [7]. Among them, leptin is a cytokine-like peptide composed of 167 amino acids, which is secreted by white adipose tissue, which regulates food intake and energy metabolism by binding to hypothalamic receptors [16]. Ghrelin and leptin can regulate the expression of feeding-promoting neuropeptides and feeding-inhibiting neuropeptides in the arcuate nucleus of the hypothalamus after binding with receptors and regulate the AMPK signaling pathway. AMPK is a heterotrimeric complex activated by upstream kinases in the AMPK signaling pathway [17]. Ghrelin can also inhibit the phosphorylation of acetyl CoA carboxylase (ACC) and reduce the expression of malonyl CoA by activating AMPK activity in the ventromedial hypothalamus (VMH) so as to inhibit carnitine palmitoyltransferase 1 (CPT-1) and finally increase the oxidation of fatty acids and increase food intake [18]. Leptin can also inhibit the activity of AMPK by activating mammalian target of rapamycin complex 1 (mTORC1), and it can also act downstream of the AMPK signaling pathway, increase ACC [19] and malonyl CoA, and inhibit food intake [19,20]. Moreover, neuropeptide Y (NPY) and agouti-related protein (AgRP) in the arcuate nucleus of the hypothalamus, as well as neuropeptide suppressants, proopiomelanocortin (POMC), and cocaine- and amphetamine-regulated transcription peptide (CART), were involved in the regulation of AMPK [21].

*Eothenomys milletus* belong to the genus *Eothenomys* (Microtinae, Cricetidae, Rodentia), which is endemic to China and widely distributed in the Hengduan Mountain and its adjacent areas [22]. Our previous research results showed that their body mass, RMR, serum leptin, and hypothalamic neuropeptides have phenotypic plasticity in *E. miletus* under random feeding restriction, and the physiological indicators showed regional differences [23]. However, the role of ghrelin, leptin, and the AMPK signaling pathway in its body mass regulation under food restriction is not clear. Therefore, based on previous studies, this study showed the phenotypic changes in ghrelin, leptin, and the AMPK signaling pathway and other physiological indicators of *E. miletus* in Kunming (KM) and Dali (DL) regions under 80% food restriction. We hypothesized that *E. miletus* from different regions will also have regional adaptation differences under the condition of food restriction and that ghrelin and the AMPK pathway play an important role in energy metabolism. Our prediction is that the body mass and metabolic rate will decrease after food restriction, while the liver mass, the concentration of ghrelin, the expression of feeding-promoting neuropeptides, and the activity of AMPK will increase. On the contrary, the concentration of leptin and the expression of feeding-inhibiting neuropeptides will decrease after food restriction. The purpose of this study is to clarify the survival strategies of *E. milletus* under food shortage, which provides a theoretical basis for the study of adaptation strategies of small mammals in Yunnan Province.

## 2. Materials and Methods

### 2.1. Subjects and Experimental Design

In total, 36 *E. miletus* (♂18:♀18) specimens were captured and selected in KM (102.80° E, 24.88° N, altitude 2020 m, *n* = 18) and DL (99.90° E, 26.53° N, altitude 2590 m, *n* = 18) in a mouse cage in the winter of 2021. Annual climate patterns in Kunming and Dali are shown in Table 1. *E. miletus* were sterilized to kill fleas, then brought back to the laboratory of Yunnan Normal University, and housed singly in a transparent plastic box (L260 mm × W160 mm × H150 mm) at 25 ± 1 °C in a photoperiod of 8L:16D; food (standard mice chow pellets; produced by Kunming Medical University, Kunming, China) and water were provided ad libitum. The 36 voles used in this study were all in non-pregnant, lactating, or young individuals. All animals were acclimatized in the laboratory for 4 d, and the body mass and food intake were measured every day. The average food intake was taken as the daily intake standard of the experiment.

Animals were randomly divided into 6 groups: a control group between KM and DL (C-KM (♂3:♀3), C-DL (♂3:♀3)), an 80% food restriction 30 d group (F80%-KM (♂3:♀3), F80%-DL (♂3:♀3)), and a refeeding 30 d group (Re30d-KM (♂3:♀3), Re30d-DL (♂3:♀3)). The 80% food restriction was based on the food intake measured during the early feeding period. There was no significant difference in the animal body mass between the control group, food restriction group, and heavy-feeding group in the same area. The body mass, food intake, and RMR in *E. miletus* were measured on days 0, 30, and 60 for the corresponding group (day 0: control group; day 30: 80% food restriction group; day 60: refeeding group), respectively. After the measurement of the body mass, food intake, and RMR, an injection of pentobarbital sodium (50 mg/kg) was administered, then the animals were killed, and the serum, medial hypothalamus, and stomach were removed and stored in a freezer (−80 °C) and reserved for the subsequent determination of leptin, total ghrelin, and other indicators.

### 2.2. Measurement of Body Mass and Food Intake

The body mass was measured using an LT502 electronic balance (accurate to 0.01 g); food intake was measured using the food balance method [22].

### 2.3. Measurement of RMR

The RMR was measured using an 8-channel FMS portable respiratory metabolic measurement system (Sable Systems International, Inc., North Las Vegas, NV, USA), and the animals were fasted for 3–4 h before testing. Animals were placed in respiratory chambers (volume of respiratory chamber 1.5 L), and the air flow of the air pump was set to 200 mL/min. The experimental temperature was controlled using an SPX-300 artificial climate chamber produced by Shanghai Boxun Medical Equipment Factory (temperature fluctuation ±0.5 °C), the experimental temperature was controlled at 25 ± 0.5 °C [24], the animals were acclimatized in the metabolic chamber for 30 min in the resting state, and the RMR was measured using ExpeData software for 4 rounds of 15 min each. At the end of the experiment, the data were exported to obtain the oxygen consumption and carbon dioxide production of each animal, and the two lowest consecutive stable readings were selected to calculate the RMR [25].

### 2.4. Determination of Leptin, Total Ghrelin Concentration, AMPK Activity, Malonyl CoA Activity, and CPT-1 Activity

The collected blood was left standing in a 4 °C refrigerator for 1 h and then centrifuged at 4 °C (4000 r/min, 30 min), and the serum was sucked into a 2 mL centrifuge tube and stored in a cabinet freezer (−80 °C) and reserved. Serum leptin and total serum ghrelin were measured using enzyme-linked immunosorbent assay (ELISA) kits (Preferred Biotechnology Co., Shanghai, China; Kit No: Leptin: YX-E20012M, Ghrelin: YX-E22207M). Frozen stomach tissue was boiled for 10 min in Wve volumes of Milli-Q water and cooled on ice. Acetic acid was added to 1 M Wnal concentration, and the acidiWed tissues were homogenized. After centrifugation at 14,600× *g* for 30 min, 50 mg tissue equivalent of the supernatant was lyophilized and used to measure the ghrelin content in the stomach. The ghrelin content was measured from a sample of 0.1 mg tissue/tube, and the stomach ghrelin concentration was measured using enzyme-linked immunosorbent assay (ELISA) kits (Preferred Biotechnology Co., Shanghai, China; Kit No: YX-E22207M). AMPK activity, malonyl CoA activity, and CPT-1 activity in the hypothalamus were measured using enzyme-linked immunosorbent assay (ELISA) kits (Preferred Biotechnology Co., Shanghai, China; Kit No: AMPK: X-E22218M, Malonyl CoA: YX-E22221M, CPT-1: YX-E22225M).

### 2.5. Determination of Neuropeptide Gene Expression in the Hypothalamus

After *E. miletus* were sacrificed, the brain tissue was removed using rapid craniotomy and the hypothalamic tissue was isolated. Total RNA was isolated from the medial hypothalamus using the TRIzol Kit (Invitrogen, Carlsbad, CA, USA) according to the manufacturer’s protocol. To remove any contaminating DNA, RNA samples were treated with DNase I (Promega, Madison, WI, USA) at 37 °C for 30 min, followed by another cycle of TRIzol extraction to eliminate residual DNase I. An equal amount (3 μg) of total RNA was transcribed into first-strand cDNA for each sample using the M-MLV First Strand Kit (Invitrogen) according to the manufacturer’s instructions (Preferred Biotechnology Co., Shanghai, China; Kit No: NPY: YX-E20502M; AGRP: YX-E22211M; POMC: YX-E22212M; CART: YX-E22217M).

Primers set for β-actin and four hypothalamic genes were used for real-time qPCR (Table 2) [26]. Standard curves were constructed for each gene via serial dilutions of cDNA (1- to 26-fold dilutions). Analysis of standard curves between target genes and β-actin showed that they had a similar amplification efficiency, which ensures the validity of the comparative quantity method. Real-time qPCR was completed using the SYBR Green I qPCR kit (Invitrogen) in the ABI Prism^®^ 7000 Sequence Detection system (Applied Biosystems, Carlsbad, CA, USA). Real-time qPCR was carried out in 20 μL of a reaction agent comprising 9.5 μL of RNase-free ddH_2_O, 9.0 μL of Platinum^®^ Quantitative PCR SuperMix-UDG (including Rox), 0.5 μL of cDNA templates, 0.5 μL of a 10 μmoL/L forward primer, and 0.5 μL of a 10 μmoL/L reverse primer. Each sample was analyzed in triplicate. Thermal cycling conditions were 50 °C for 120 s, 95 °C for 120 s, 45 cycles of 95 °C for 15 s, and 60 °C for 45 s (Preferred Affinity, San Diego, CA, USA; Article No: β-Actin: 4970S).

### 2.6. Carcass Mass, Body Fat Mass and Content, Body Composition, and Digestive Tract Morphology Determination

The viscera-removed animal carcass (fat retention on the digestive tract) was weighed to obtain the wet mass of the carcass, and the carcass was dried to a constant weight in an oven at 60 °C to obtain the dry mass of the carcass. The body fat mass was measured using a Soxtec^TM^2043 extraction device: After the dried carcass was crushed and mixed using a small grinder, about 2 g of the crushed sample was weighed and the fat weight was measured using a Soxtec^TM^2043 extraction device (Flowserve, Hilleroed, Denmark). The body fat content was calculated according to the following formula: body fat content (%) = total fat weight (g)/carcass dry weight (g) × 100%. Organs, including the heart, liver, lungs, spleen, kidneys, and gonads, were carefully separated, the connective tissue and lipids adhering to the organs were removed, and the blood on the surface of the organs was blotted dry using filter paper. The organs were weighed (accurate to l mg) and wrapped in tin foil, placed in an oven, baked at 60 °C to a constant mass, and then weighed (accurate to l mg) as the dry mass of the organs. The digestive tract of the animals was removed; the stomach, small intestine, large intestine, and cecum were separated; the mesentery and connective tissue and lipids of each organ were carefully removed, flattened to the maximum length, and not stretched; and then the length of each part was measured (± 1 mm). After each organ was dried on filter paper, it was placed on tin platinum paper, its wet mass with contents was weighed, and the stomach was used for the content of ghrelin, so only the wet mass with contents was available. The organs were then dissected longitudinally, washed using physiological saline to remove the contents, placed on filter paper to absorb the surface water, and weighed as wet mass with no content of the organs. Each organ was wrapped in tin foil after weighing. The organs were placed in an oven, dried at 60 °C to a constant mass, and weighed as the dry mass of the organs.

### 2.7. Data Analysis

Data were analyzed using SPSS 22.0 software (IBM, Armonk, NY, USA). Before all statistical analyses, data were examined for normality and homogeneity of variance using Kolmogorov–Smirnov and Levene tests, respectively. Differences in physiological indicators between the different sexes of *E. miletus* in both regions were not significant, so all data were combined and counted. Two-way ANOVA (region × food restriction) was performed to analyze the differences in body mass between different groups, and two-way ACNOVA was performed to analyze the differences in other physiological indicators, with the body mass or carcass mass as a covariate, followed by Tukey’s honestly significant difference post hoc tests. Pearson correlation analysis was performed to analyze the relationship between the two indicators. Results were expressed by the mean ± SEM, and *p* < 0.05 meant a significant difference.

## 3. Results

### 3.1. Body Mass, RMR, and Body Composition

The region and food restriction had significant effects on the body mass of *E. miletus* (region: F_1,30_ = 5.70, *p* = 0.02; food restriction: F_2,30_ = 22.52, *p* < 0.01; region × food restriction: F_2,30_ = 0.03, *p* = 0.97). The region and food restriction had significant effects on the RMR (region: F_1,30_ = 11.22, *p* < 0.01; food restriction: F_2,30_ = 4.25, *p* = 0.02; region × food restriction: F_2,30_ = 2.03, *p* = 0.15). Although food restriction reduced the RMR in DL, the effect of food restriction on the RMR of the three groups in DL was not significant. The body mass in KM was significantly higher than that in DL, but the RMR in KM was significantly lower than that in DL. The body mass in KM decreased by 14.51% (Figure 1a), and the RMR decreased by 12.46% (Figure 1b) compared to the control group; the body mass in DL decreased by 14.81% (Figure 1a), and the RMR decreased by 3.84% (Figure 1b) compared to the control group. After refeeding, the RMR in both regions returned to the level of the control group (Figure 1a,b).

Food restriction had a significant impact on the body fat mass and fat content, while the region had a significant impact on the fat mass and fat content, but the interaction between them was not significant. Food restriction and the region had no effect on the wet and dry mass of the carcass (Table 3). Food restriction had a significant effect on the wet and dry mass of the liver in *E. miletus* (liver wet mass: F_2,30_ = 4.74, *p* = 0.02; liver dry mass: F_2,30_ = 3.78, *p* = 0.04, Table 4). The region had a significant effect on the small intestine wet mass with content, the large intestine length, and the cecum length and cecum dry mass with content (small intestine wet mass with content: F_1,30_ = 24.21, *p* < 0.01; large intestine length: F_1,30_ = 6.10, *p* = 0.02; Cecum length: F_1,30_ = 5.42, *p* = 0.03; cecum dry mass with content: F_1,30_ = 6.17, *p* = 0.02; Table 5). The interaction between region and food restriction had a significant impact on the small intestine dry mass with content and the length of the cecum (small intestine dry mass with content: F_2,30_ = 7.38, *p* < 0.01; cecum length: F_2,30_ = 3.79, *p* = 0.04; Table 5). Other body composition and digestive tract data had no significant changes. After 30 days of refeeding, the wet and dry mass of the liver of *E. miletus* could return to the level of the control group (Table 4).

### 3.2. Changes in Neuropeptide Gene Expressions in the Hypothalamus

Food restriction had a significant effect on the gene expression of NPY, AgRP, POMC, and CART in the hypothalamus of *E. miletus* (NPY: F_2,30_ = 13.64, *p* < 0.01; AgRP: F_2,30_ = 17.55, *p* < 0.01; POMC: F_2,30_ = 9.67, *p* < 0.01; CART: F_2,30_ = 7.82, *p* < 0.01), and the influence of the region on the gene expression of AgRP, POMC, and CART in *E. miletus* was significant (AgRP: F_1,30_ = 39.49, *p* < 0.01; POMC: F_1,30_ = 6.87, *p* = 0.01; CART: F_1,30_ = 34.78, *p* < 0.01). The interaction between region and 80% food restriction did not affect the gene expression of NPY, AgRP, POMC, and CART (NPY: F_2,30_ = 0.67, *p* = 0.52; AgRP: F_2,30_ = 0.37, *p* = 0.69; POMC: F_2,30_ = 0.60, *p* = 0.56; CART: F_2,30_ = 0.60, *p* = 0.56). The gene expression of AgRP in DL was higher than that in KM, but the gene expression of POMC and CART was lower in KM than that in DL. After 30 days of food restriction, NYP gene expression in the hypothalamus in KM increased by 33.95% (Figure 2a), AgRP expression increased by 31.98% (Figure 2b), and POMC and CART expression decreased by 21.99% (Figure 2c) and 20.47% (Figure 2d), respectively. However, NYP gene expression in the hypothalamus in DL increased by 20.33% (Figure 2a), AgRP expression increased by 26.11% (Figure 2b), and POMC and CART expression decreased by 28.45% (Figure 2c) and 32.58% (Figure 2d), respectively. After 30 days of refeeding, the gene expression levels of NPY, AgRP, and POMC in the two regions returned to the level of the control group (Figure 2a–d).

### 3.3. Changes in Serum Leptin, Serum Ghrelin, and Stomach Ghrelin Concentration

Food restriction had significant effects on the concentration of serum leptin, serum ghrelin, and stomach ghrelin (leptin: F_2,30_ = 12.46, *p* < 0.01; serum ghrelin: F_2,30_ = 19.34, *p* < 0.01; stomach ghrelin: F_2,30_ = 9.66, *p* < 0.01). The region led to significant differences in the concentration of serum leptin, serum ghrelin, and stomach ghrelin of *E. miletus* (leptin: F_1,30_ = 50.38, *p* < 0.01; serum ghrelin: F_1,30_ = 4.573, *p* < 0.05; stomach ghrelin: F_1,30_ = 19.57, *p* < 0.01). The interaction of region and food restriction affected the concentration of serum leptin (leptin: F_2,30_ = 3.44, *p* = 0.05), but it did not affect the concentration of ghrelin in the serum and stomach (serum ghrelin: F_2,30_ = 1.87, *p* = 0.17; stomach ghrelin: F_2,30_ = 0.73, *p* = 0.49). The concentrations of serum leptin and stomach ghrelin in KM were higher than those in DL, and the serum ghrelin concentration in DL was higher than that in KM. After 30 days of food restriction, the concentration of serum ghrelin and stomach ghrelin increased, while the concentration of serum leptin decreased. The content of serum ghrelin and stomach ghrelin increased by 47.83% (Figure 3b) and 34.39% (Figure 3c) in KM, respectively, and the concentration of serum leptin decreased by 14.49% (Figure 3a). In DL, the concentration of ghrelin in the serum and stomach increased by 70.86% (Figure 3b) and 35.29% (Figure 3c), respectively, and the concentration of leptin in the serum decreased by 31.91% (Figure 3a). After refeeding, the concentration of serum leptin, serum ghrelin, and stomach ghrelin returned to the level of the control group (Figure 3a–c).

### 3.4. Changes in AMPK, Malonyl CoA, and CPT-1 Activities in the Hypothalamus

Food restriction had significant effects on the activities of AMPK, malonyl CoA, and CPT-1 (AMPK: F_2,30_ = 19.79, *p* < 0.01; malonyl CoA: F_2,30_ = 17.94, *p* < 0.01; CPT-1: F_2,30_ = 20.17, *p* < 0.01), and the effect of region on the activities of AMPK, malonyl CoA, and CPT-1 in the hypothalamus of *E. miletus* was significantly different (AMPK: F_1,30_ = 247.66, *p* < 0.01; malonyl CoA: F_1,30_ = 28.47, *p* < 0.01; CPT-1: F_1,30_ = 284.24, *p* < 0.01). The interaction of region and food restriction did not affect the activities of AMPK and malonyl CoA in the hypothalamus (AMPK: F_2,30_ = 0.28, *p* = 0.76; malonyl CoA: F_2,30_ = 0.51, *p* = 0.61), but it affected CPT-1 activity (CPT-1: F_2,30_ = 3.73, *p* = 0.04). AMPK, malonyl CoA, and CPT-1 activities in KM were higher than those in DL. Food restriction reduced the malonyl CoA and CPT-1 activities in both regions but increased AMPK activity. Among them, the AMPK activity in KM increased by 32.81% (Figure 4a), and the malonyl CoA and CPT-1 activities decreased by 9.97% (Figure 4b) and 18.09% (Figure 4c), respectively; the AMPK activity in DL increased by 71.79% (Figure 4a), and the malonyl CoA and CPT-1 activities decreased by 17.37% (Figure 4b) and 10.60% (Figure 4c), respectively. After refeeding, the activities of AMPK and malonyl CoA returned to the level of the control group (Figure 4a–c).

### 3.5. Correlation Analysis

The stomach ghrelin concentration was positively correlated with NPY expression (Figure 5d), AgRP expression (Figure 5e), serum ghrelin concentration (Figure 5h), and AMPK activity (Figure 5j) and negatively correlated with the body mass (Figure 5a), RMR (Figure 5b), food intake (Figure 5c), POMC expression (Figure 5f), CART expression (Figure 5g), serum leptin concentration (Figure 5i), malonyl CoA activity (Figure 5k), and CPT-1 activity (Figure 5l). The concentration of serum leptin was positively correlated with the body mass (Figure 6a), RMR (Figure 6b), food intake (Figure 6c), POMC gene expression (Figure 6f), CART gene expression (Figure 6g), malonyl CoA activity (Figure 6j), and CPT-1 activity (Figure 6k) and negatively correlated with NPY gene expression (Figure 6d), AgRP gene expression (Figure 6e), serum ghrelin concentration (Figure 6h), and AMPK activity (Figure 6i). The serum ghrelin concentration was positively correlated with NPY gene expression (Figure 7d), AgRP gene expression (Figure 7e), and AMPK activity (Figure 7h) and negatively correlated with the body mass (Figure 7a), RMR (Figure 7b), food intake (Figure 7c), POMC gene expression (Figure 7f), CART gene expression (Figure 7g), malonyl CoA activity (Figure 7i), and CPT-1 activity (Figure 7j).

## 4. Discussion

### 4.1. Body Mass, RMR, and Body Composition

Animals follow a series of strategies to deal with the shortage of food resources, and previously, research showed that the changes in the body mass and thermogenic capacity are affected by the degree of food restriction [23]. For example, starvation reduced the body mass and RMR in *Rattus norvegicus* [27]. The body mass and RMR in *E. miletus* returned to the control level after refeeding [23]. However, for *Microtus ochrogaster* and *Clethrionomys rutilus*, these species increased their body mass after refeeding [28]. In this study, results showed that the body mass and RMR of *E. miletus* decrease under food restriction, suggesting that *E. miletus* could survive by reducing the body mass and RMR under the condition of insufficient food resources, which is consistent with the results on *Mesocricetus auratus* and *Phodopus sungorus* [29,30]. Moreover, the body mass and RMR returned to the control level after refeeding, indicating that *E. miletus* has strong phenotypic plasticity. Phenotypic changes were also found in internal organs, which may reflect their functional adaptation [31]. Food restriction decreased the wet and dry mass of the liver in *E. miletus,* which reflected that food restriction reduces the thermogenic capacity of *E. miletus* to maintain their survival [32]. Furthermore, in our results, the *E. miletus* in DL lost more body mass than that in KM after food restriction, which indicated that the food situation in winter in DL is poor and *E. miletus* can adjust their body composition in time to deal with the changes in food resources [33].

### 4.2. Ghrelin, Leptin, AMPK Activity, Malonyl CoA, CPT-1, and Hypothalamic Neuropeptide Expression

Studies have shown that food restriction increases ghrelin secretion, while refeeding inhibits ghrelin secretion [34]. In this study, stomach and serum ghrelin concentrations increased but the serum leptin concentration decreased after food restriction. Results also showed that the ghrelin concentration in the stomach after food restriction was higher than that in the serum, and the possible reasons were that food restriction was relatively long, and when *E. miletus* were threatened by hunger, ghrelin was secreted in large quantities from the stomach fundus. Therefore, more ghrelin can be detected in the stomach after food restriction. After refeeding, ghrelin decreased with stomach fill and leptin increased with increased fat mass. Moreover, the levels of leptin and ghrelin in KM were higher than those in DL, which can be understood as a better food living environment in KM in winter; *E. miletus* in KM would secrete more ghrelin and leptin to regulate their body mass and energy expenditure under food shortage [23].

The exogenous activation of hypothalamic AMPK increases food intake, while exogenous inhibition reduces food intake in mammals [7,9]. Food restriction led to a large amount of ghrelin release, and it was expected that ghrelin would inhibit the phosphorylation of acetyl coenzyme A carboxylase (ACC) after inducing AMPK activation, thereby reducing malonyl CoA levels, thereby reducing CPT-1 release. On the contrary, serum leptin was secreted in proportion to the fat level, and leptin inhibited hypothalamic AMPK activity, thereby limiting food intake [17]. In this study, food restriction increased the activity of AMPK in the hypothalamus but decreased the activities of malonyl CoA and CPT-1.

Food restriction reduced the body fat mass, and the reduction in the body fat mass decreased the leptin level [35] and allowed ghrelin to stimulate the reduction in malonyl CoA induced by GHS-R1a-AMPK to activate CPT-1a and CPT-1c to activate NPY/AgRP neurons [20]. Studies have shown that food restriction increases the gene expression of AgRP and NPY [36,37], while gene deletion of AgRP neurons leads to hunger and body mass loss [38,39]. Moreover, AMPK is essential for energy homeostasis regulation and glucose sensing by POMC and AgRP neurons, and if AMPK activity reduces, AgRP neurons are unable to fully promote eating and cause body mass loss [40]. In this study, food restriction significantly affected the expression of neuropeptides in the hypothalamus, which returned to the level of the control group after refeeding, indicating that *E. miletus* has certain phenotypic plasticity. NPY and AgRP gene expression in DL was higher than that in KM, while POMC and CART gene expression in KM was higher than that in DL. Our explanation for this is that DL has a high altitude and poor food resources in winter; therefore, in the face of food shortages, *E. miletus* in DL quickly release hunger signals to promote appetite and start for aging to survive.

It is also worth noting that ghrelin and leptin did not work separately. Researchers found that ghrelin levels increase in rats under food restriction, which confirms the negative regulation of circulating leptin on the plasma ghrelin concentration. By studying the effect of peripheral injection of leptin on rats with food restriction, it is concluded that moderate increase in leptin can not only act on the hypothalamus but also cause satiety in the periphery by directly reducing the ghrelin signal in the gastrointestinal tract [41]. In this study, the stomach ghrelin concentration was positively correlated with the serum ghrelin concentration and AMPK activity and negatively correlated with the serum leptin concentration. Serum leptin was negatively correlated with the serum ghrelin concentration and AMPK activity, and the serum ghrelin concentration was positively correlated with AMPK activity. This is because negative regulation between ghrelin and leptin induces changes in appetite and/or energy expenditure to maintain the energy balance; moreover, some studies have shown that AMPK activation is the key mechanism to maintain appetite and the joint action of ghrelin and leptin [18].

In addition to the hypothalamus, vagal afferent neurons also express LRB receptors [42,43]. The binding of leptin to LRB receptors stimulates the autophosphorylation of Janus-activated kinase 2 (JAK2), which in turn activates STAT3 [44,45,46]. Similar to the LRB receptor, the functional ghrelin receptor GHS-R1a is synthesized in the nodal ganglion (NG) and transmitted to the vagal afferent terminals. Ghrelin secreted from the stomach may interact with GHS-R1a expressed at these terminals, and the generated signals may be transmitted to the hypothalamus through the nucleus tractus solitarius [47]. Feeding studies have shown that ghrelin [48] and leptin [49] regulate food intake through the vagal afferent pathway. Other studies have shown that the endogenous cannabinoid system is equally important for ghrelin to stimulate AMPK activity and food intake. Ghrelin stimulates the biosynthesis of 2-arachidonic acid glycerin (an endogenous cannabinoid) in PVN fine-cell neurons to inhibit the release of the excitatory neurotransmitter glutamate from presynaptic axons that stimulate PVN neurons, thereby promoting food intake [50]. Cytokine signal transduction inhibitor 3 (SOCS3) plays a key role in the regulation of leptin signaling by inhibiting the phosphorylation of JAK2 and downstream STAT3 activated by leptin [51]. In this way, SOCS3 negatively regulates hypothalamic leptin signaling and plays an important role in leptin resistance. Ghrelin induces leptin resistance through the SOCS3 pathway. Ghrelin inhibits the phosphorylation of STAT3 and JAK2 in leptin-stimulated NG neurons in *Rattus norvegicus*. A combined injection of cholecystokinin and leptin reduced food intake by increasing CART/TRH and reducing the phosphorylation of hypothalamic AMPK [52]. In this study, we also speculated that ghrelin and leptin might work together to maintain the AMPK signaling function to regulate energy metabolism in *E. miletus*.

## 5. Conclusions

In conclusion, food restriction reduced the body mass and RMR in *E. miletus*; increased the concentration of ghrelin in the serum and stomach; reduced the concentration of serum leptin, which would activate the hypothalamic AMPK signal pathway; and finally reduced CPT-1. Changes in ghrelin and leptin eventually led to an increase in NPY and AgRP gene expression and an increase in food intake. The ghrelin- and leptin-combined AMPK signaling pathway in the hypothalamus could play an important role in regulating feeding when food resources are restored so as to ensure that *E. miletus* maintain energy homeostasis. Moreover, body mass regulation in *E. miletus* in DL was more sensitive, which might be due to poor food resources and cold temperatures in winter in DL (Figure 8).

## Figures and Tables

**Figure 1 animals-12-03333-f001:**
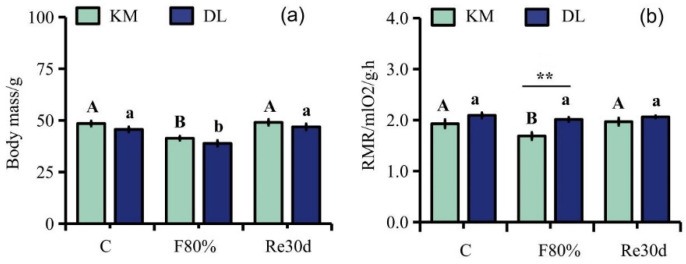
Changes in body mass (**a**) and RMR (**b**) in *E. miletus* under food restriction. Different letters denote significant differences among treatments in the same region, capital letters refer to the Kunming region, and lowercase letters refer to the Dali region. ** *p* < 0.01 Kunming versus Dali on the same day.

**Figure 2 animals-12-03333-f002:**
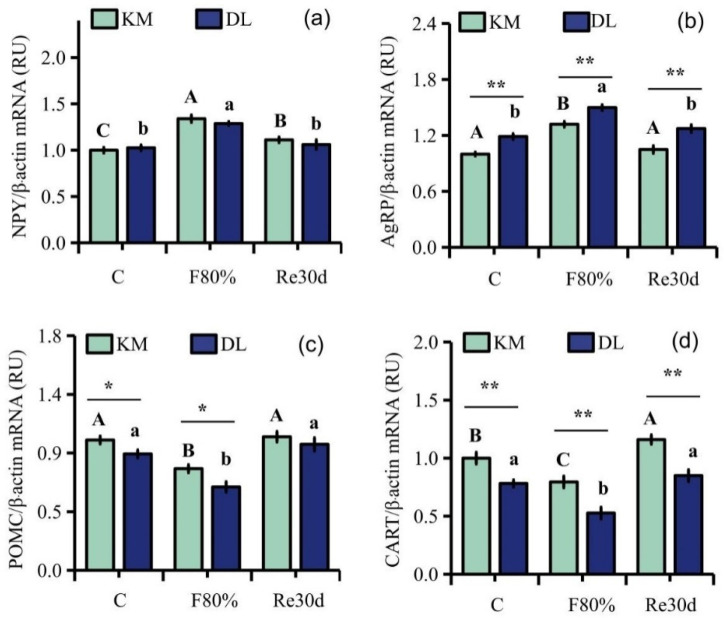
Gene expression levels of NPY (**a**), AgRP (**b**), POMC (**c**), and CART (**d**) in the hypothalamus of *E. miletus* under food restriction. Different letters denote significant differences among treatments in the same region, capital letters refer to the Kunming region, and lowercase letters refer to the Dali region. * *p* < 0.05, ** *p* < 0.01 Kunming versus Dali on the same day.

**Figure 3 animals-12-03333-f003:**
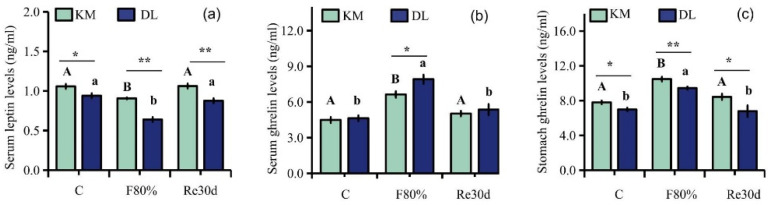
Serum leptin (**a**), serum ghrelin (**b**), and stomach ghrelin concentrations (**c**) in *E. miletus* under food restriction. Different letters denote significant differences among treatments in the same region, capital letters refer to the Kunming region, and lowercase letters refer to the Dali region. * *p* < 0.05 and ** *p* < 0.01 Kunming versus Dali on the same day.

**Figure 4 animals-12-03333-f004:**
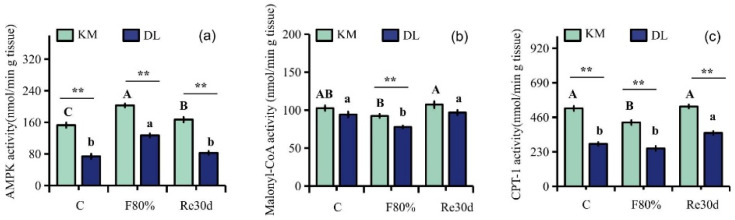
Activity diagram of AMPK (**a**), malonyl CoA (**b**), and CPT-1 (**c**) in *E. miletus* under food restriction. Different letters denote significant differences among treatments in the same region, capital letters refer to the Kunming region, and lowercase letters refer to the Dali region. ** *p* < 0.01 Kunming versus Dali on the same day.

**Figure 5 animals-12-03333-f005:**
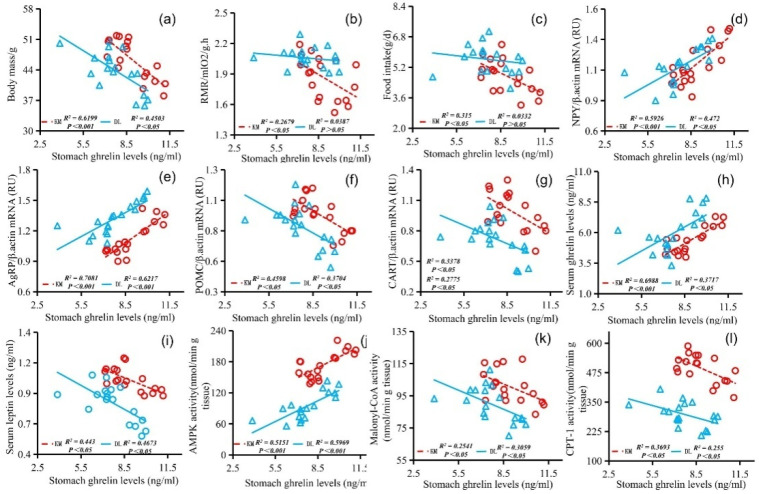
Correlation diagram between stomach ghrelin concentration and body mass (**a**), RMR (**b**), food intake (**c**), NPY gene expression (**d**), AgRP gene expression (**e**), POMC gene expression (**f**), CART gene expression (**g**), serum ghrelin concentration (**h**), serum leptin concentration (**i**), AMPK activity (**j**), malonyl CoA activity (**k**), and CPT-1 activity (**l**) in *E. miletus* under food restriction.

**Figure 6 animals-12-03333-f006:**
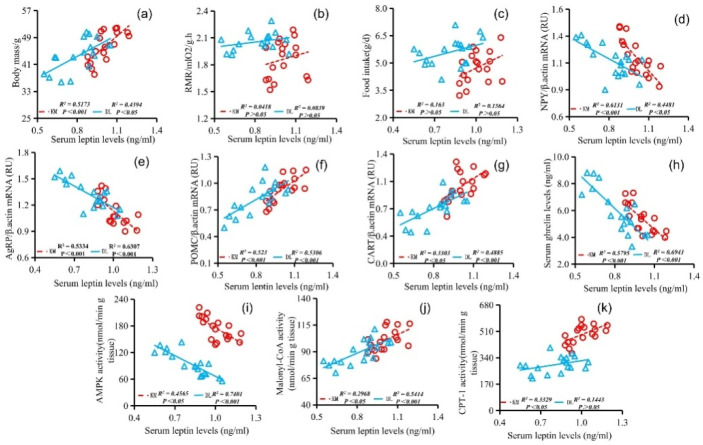
Correlation between serum leptin concentration and body mass (**a**), RMR (**b**), food intake (**c**), NPY gene expression (**d**), AgRP gene expression (e), POMC gene expression (**f**), CART gene expression (**g**), serum ghrelin concentration (**h**), AMPK activity (**i**), malonyl CoA activity (**j**), and CPT-1 activity (**k**) in *E. miletus* under food restriction.

**Figure 7 animals-12-03333-f007:**
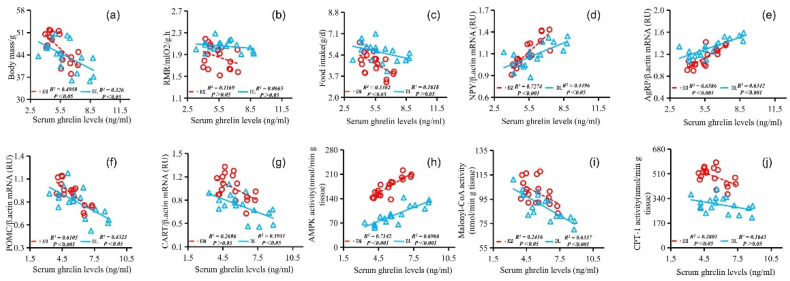
Correlation between serum ghrelin concentration and body mass (**a**), RMR (**b**), food intake (**c**), NPY gene expression (**d**), AgRP gene expression (**e**), POMC gene expression (**f**), CART gene expression (**g**), AMPK activity (**h**), malonyl CoA activity (**i**), and CPT-1 activity (**j**) in *E. miletus* under food restriction.

**Figure 8 animals-12-03333-f008:**
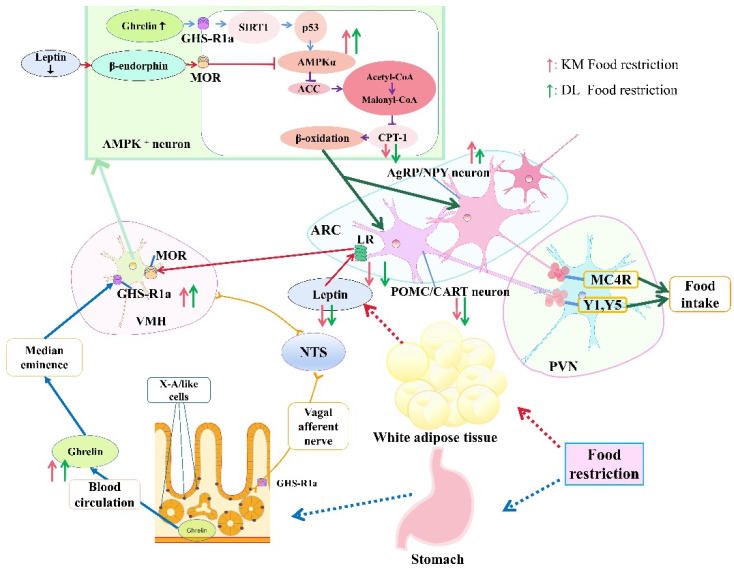
Roles of ghrelin and leptin in body mass regulation under food restriction based on the AMPK pathway in *E. miletus* from Kunming and Dali regions. The height of the arrow represents the change in the indicator between the two regions. The higher the arrow is, the greater the change is.

**Table 1 animals-12-03333-t001:** Altitude, coordinates, temperature, sunshine, precipitation, and vegetation types in Kunming and Dali regions.

	Kunming (KM)	Dali (DL)
Longitude	102°51′57″ E	99°75′03″ E
Latitude	24°52′4″ N	26°43′95″ N
Altitude	2020 m	2590 m
Minimum temperature	−1 °C	−5 °C
Maximum temperature	30.1 °C	30 °C
Annual average temperature	16.5 °C	7 °C
Rainfall	1026.1 mm	657.5 mm
Sunshine	198.4–343.1 h	191.2–220.6 h
Vegetation type	Small leaved shrub	Evergreen broad-leaf forest

**Table 2 animals-12-03333-t002:** Gene-specific primers used for real-time qPCR.

Primer	Oligonuncleotide Sequence (5′ to 3′)	Product Size (bp)
NPY (forward)	TGGACTGACCCTCGCTCTAT	162
NPY (reverse)	GTGTCTCAGGGCTGGATCTC	
AgRP (forward)	AGAGTTCTCAGGTCTAAGTCT	187
AgRP (reverse)	CTTGAAGAAGCGGCAGTAGCACGT	
POMC (forward)	CCTGTGAAGGTGTACCCAATGTC	240
POMC (reverse)	CACGTTCTTGATGATGGCGTTC	
CART (forward)	AGAAGAAGTACGGCCAAGTCC	55
CART (reverse)	CACACAGCTTCCCGATCC	
β-actin (forward)	GAGAGGGAAATCGTGCGTGAC	170
β-actin (reverse)	CATCTGCTGGAAGGTGGACA	

**Table 3 animals-12-03333-t003:** Carcass mass, body fat mass, and content of *E. miletus* in different regions under food restriction.

Parameters	KM			DL			Statistical Summary					
	C-KM	F_80%_-KM	Re_30d_-KM	C-DL	F_80%_-DL	Re_30d_-DL	Region		Food Restriction		Region × Food Restriction	
	(*n* = 6)	(*n* = 6)	(*n* = 6)	(*n* = 6)	(*n* = 6)	(*n* = 6)	F_1,30_	*p*	F_2,30_	*p*	F_2,30_	*p*
Carcass wet mass (g)	32.88 ± 1.27	26.24 ± 1.47	30.59 ± 1.95	29.52 ± 1.43	27.72 ± 1.39	30.59 ± 3.13	0.140	0.711	0.182	0.835	0.856	0.435
Carcass dry mass (g)	19.08 ± 1.14	14.45 ± 1.32	16.76 ± 1.96	15.69 ± 1.28	15.67 ± 1.29	17.41 ± 2.52	0.139	0.712	1.165	0.326	1.144	0.332
Body fat mass (g)	8.03 ± 0.47	3.36 ± 0.31	7.16 ± 0.87	6.28 ± 0.51	3.34 ± 0.29	7.22 ± 1.00	6.712	<0.05	168.320	<0.01	0.473	0.628
Body fat content (%)	0.42 ± 0.01	0.23 ± 0.01	0.43 ± 0.00	0.40 ± 0.00	0.21 ± 0.01	0.42 ± 0.00	11.298	<0.01	540.402	<0.01	0.705	0.503

**Table 4 animals-12-03333-t004:** Body composition of *E. miletus* in different regions under food restriction.

Parameters	KM			DL			Statistical Summary					
	C-KM	F_80%_-KM	Re_30d_-KM	C-DL	F_80%_-DL	Re_30d_-DL	Region		Food restriction		Region × Food Restriction	
	(*n* = 6)	(*n* = 6)	(*n* = 6)	(*n* = 6)	(*n* = 6)	(*n* = 6)	F_1,30_	*p*	F_2,30_	*p*	F_2,30_	*p*
Heart wet mass (g)	0.25 ± 0.01	0.21 ± 0.02	0.25 ± 0.02	0.25 ± 0.02	0.23 ± 0.02	0.27 ± 0.02	0.808	0.376	0.516	0.602	0.093	0.911
Liver wet mass (g)	3.15 ± 0.37	1.65 ± 0.16	2.75 ± 0.09	2.39 ± 0.20	1.42 ± 0.30	2.58 ± 0.12	2.106	0.158	4.735	<0.05	0.930	0.406
Spleen wet mass (g)	0.11 ± 0.01	0.08 ± 0.01	0.09 ± 0.01	0.11 ± 0.03	0.07 ± 0.01	0.09 ± 0.01	0.122	0.729	2.964	0.067	0.067	0.935
Lungs wet mass (g)	0.36 ± 0.02	0.27 ± 0.03	0.29 ± 0.01	0.28 ± 0.03	0.28 ± 0.03	0.29 ± 0.03	0.394	0.535	0.565	0.574	1.504	0.239
Kidney wet mass (g)	0.39 ± 0.01	0.37 ± 0.02	0.32 ± 0.03	0.39 ± 0.02	0.37 ± 0.03	0.39 ± 0.03	1.556	0.222	1.446	0.252	1.445	0.252
Heart dry mass (g)	0.06 ± 0.01	0.05 ± 0.003	0.07 ± 0.01	0.07 ± 0.01	0.06 ± 0.01	0.07 ± 0.01	2.968	0.096	1.117	0.341	0.369	0.695
Liver dry mass (g)	1.06 ± 0.15	0.51 ± 0.04	1.00 ± 0.08	0.73 ± 0.06	0.58 ± 0.03	0.92 ± 0.06	1.205	0.281	3.777	<0.05	3.284	0.052
Spleen dry mass (g)	0.03 ± 0.002	0.02 ± 0.002	0.03 ± 0.003	0.03 ± 0.002	0.02 ± 0.002	0.02 ± 0.004	0.005	0.943	1.688	0.203	0.128	0.880
Lungs dry mass (g)	0.10 ± 0.01	0.06 ± 0.01	0.08 ± 0.003	0.08 ± 0.01	0.07 ± 0.01	0.09 ± 0.01	0.244	0.652	0.998	0.381	2.036	0.149
Kidney dry mass (g)	0.13 ± 0.01	0.11 ± 0.01	0.11 ± 0.01	0.14 ± 0.01	0.12 ± 0.01	0.13 ± 0.01	2.996	0.094	2.905	0.071	0.020	0.980

**Table 5 animals-12-03333-t005:** Digestive tract changes in *E. miletus* in different regions under food restriction.

Parameters	KM			DL			Statistical Summary					
	C-KM	F_80%_-KM	Re_30d_-KM	C-DL	F_80%_-DL	Re_30d_-DL	Region		Food Restriction		Region × Food Restriction	
	(*n* = 6)	(*n* = 6)	(*n* = 6)	(*n* = 6)	(*n* = 6)	(*n* = 6)	F_1,30_	*p*	F_2,30_	*p*	F_2,30_	*p*
Stomach wet mass with content (g)	0.50 ± 0.09	0.46 ± 0.07	0.64 ± 0.09	0.39 ± 0.05	0.36 ± 0.04	0.46 ± 0.09	3.745	0.063	1.211	0.312	0.221	0.803
Large intestine wet mass with content (g)	0.68 ± 0.07	0.46 ± 0.06	0.58 ± 0.03	0.63 ± 0.08	0.60 ± 0.04	0.84 ± 0.19	1.262	0.271	1.258	0.299	1.415	0.259
Small intestine wet mass with content (g)	1.77 ± 0.05	1.69 ± 0.18	1.48 ± 0.09	1.68 ± 0.11	1.66 ± 0.09	1.53 ± 0.12	0.443	0.511	2.001	0.153	0.231	0.795
Cecum wet mass with content (g)	1.60 ± 0.14	2.30 ± 0.33	1.55 ± 0.25	1.53 ± 0.10	2.50 ± 0.15	2.05 ± 0.34	0.025	0.875	1.147	0.332	0.978	0.388
Stomach wet mass with no content (g)	0.22 ± 0.03	0.28 ± 0.07	0.19 ± 0.03	0.22 ± 0.02	0.19 ± 0.01	0.19 ± 0.03	0.125	0.726	2.168	0.133	1.190	0.319
Large intestine wet mass with no content (g)	0.40 ± 0.02	0.28 ± 0.02	0.40 ± 0.05	0.41 ± 0.05	0.33 ± 0.01	0.46 ± 0.05	1.435	0.241	2.001	0.153	0.306	0.739
Small intestine wet mass with no content (g)	1.01 ± 0.03	0.92 ± 0.06	0.99 ± 0.05	0.67 ± 0.07	0.70 ± 0.06	0.76 ± 0.07	24.210	<0.001	0.467	0.632	0.492	0.617
Cecum wet mass with no content (g)	0.46 ± 0.01	0.39 ± 0.06	0.36 ± 0.02	0.44 ± 0.02	0.45 ± 0.03	0.44 ± 0.02	1.980	0.170	1.092	0.349	1.617	0.216
Large intestine length (cm)	20.77 ± 0.39	19.77 ± 1.12	21.48 ± 1.48	23.67 ± 0.42	20.10 ± 0.25	22.47 ± 0.67	6.099	<0.05	0.469	0.631	1.368	0.271
Small intestine length (cm)	36.72 ± 1.52	34.77 ± 2.38	35.43 ± 0.81	36.93 ± 1.67	37.52 ± 1.20	36.13 ± 1.32	1.194	0.284	0.305	0.740	0.358	0.702
Cecum length (cm)	11.10 ± 0.93	9.78 ± 0.16	12.63 ± 0.99	11.18 ± 1.35	13.65 ± 0.87	12.22 ± 0.56	5.422	<0.05	2.015	0.152	3.789	<0.05
Large intestine dry mass with content (g)	0.08 ± 0.004	0.06 ± 0.004	0.09 ± 0.02	0.08 ± 0.01	0.07 ± 0.003	0.09 ± 0.02	0.642	0.429	0.572	0.571	0.173	0.842
Small intestine dry mass with content (g)	0.20 ± 0.01	0.15 ± 0.01	0.16 ± 0.02	0.12 ± 0.01	0.15 ± 0.01	0.18 ± 0.01	1.735	0.198	0.361	0.700	7.383	<0.05
Cecum dry mass with content (g)	0.08 ± 0.003	0.07 ± 0.01	0.07 ± 0.005	0.08 ± 0.003	0.09 ± 0.01	0.10 ± 0.01	6.174	<0.05	0.457	0.638	3.220	0.055

## Data Availability

The raw data are uploaded as an attachment.

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
