# Peer review of "Roles of Ghrelin and Leptin in Body Mass Regulation under Food Restriction Based on the AMPK Pathway in the Red-Backed Vole, Eothenomys miletus, from Kunming and Dali Regions"

_animals, 2022, doi:10.3390/ani12233333_

Round 1

Reviewer 1 Report

In this manuscript, author studied the roles of ghrelin and leptin on body mass regulation under food restriction based on AMPK pathway in Eothenomys miletus from Kunming and Dali regions, which is important for understanding how small mammals face the stress of food shortage inhabit seasonally changing environments in winter. Furthermore, the methods also provided sufficient details and fairly standard in the field. It is suitable for publication in this journal. However, the language of the article needs polishing, and there are some small problems that need to be improved. I also give some minor suggestions for improvement in the following.

1. Articles need further enhance preferably by native English speakers.

2. The introduction needs to add the hypothesis and prediction of the study

3. Add gender ratios for voles that were used in experiments.

4. Why choose 80% food restriction

5. The use of acronyms is inconsistent.

6. Why not 12:12 light cycle but 8:16? 

7. Species need to be written in Latin and italics.

8. Please delete the first paragraph in part of 3.1

9. There seems to be some mistakes in the format of the references,which need to be further proofread and improved.

10. What does the height of the shear head in Figure 8 mean.

Author Response

In this manuscript, author studied the roles of ghrelin and leptin on body mass regulation under food restriction based on AMPK pathway in Eothenomys miletus from Kunming and Dali regions, which is important for understanding how small mammals face the stress of food shortage inhabit seasonally changing environments in winter. Furthermore, the methods also provided sufficient details and fairly standard in the field. It is suitable for publication in this journal. However, the language of the article needs polishing, and there are some small problems that need to be improved. I also give some minor suggestions for improvement in the following.

  1. Articles need further enhance preferably by native English speakers.

Response: English usage in the draft was corrected by Pro. Burkart Engesser at Historisches Museum Basel, Switzerland.

  1. The introduction needs to add the hypothesis and prediction of the study

Response: We have added the hypothesis and prediction to the manuscript according to the Reviewer’s comments. We hypothesized that E. miletus from different regions will also have regional adaptation differences under the condition of food restriction, ghrelin and AMPK pathway play an important role in energy metabolism. Our prediction is that the body mass and metabolic rate will decrease after food restriction, and the mass of liver and other energy supplying organs, the content of ghrelin, the expression of feeding promoting neuropeptides, and the activity of AMPK will increase. On the contrary, the content of leptin and the expression of feeding inhibiting neuropeptides will decrease after food restriction.

  1. Add gender ratios for voles that were used in experiments.

Response: We have added the reference to the manuscript according to the Reviewer’s comments, gender ratios was ♂18: ♀18.

  1. Why choose 80% food restriction?

Response: This is because several food restriction groups of different degrees were set at the same time, 80% limit can make significant differences in various physiological indicators of E. miletus, and there is no individual death. If the food restriction is reduced to 70%, some individuals died.

  1. The use of acronyms is inconsistent.

Response: We have checked the acronyms and made some revision according to the Reviewer’s comments.

  1. Why not 12:12 light cycle but 8:16? 

Response: Our animals were collected in winter, and the sunshine time was short in winter, so we simulated short light under laboratory conditions, and chose 8:16.

  1. Species need to be written in Latin and italics.

Response: We have made some revision according to the Reviewer’s comments.

  1. Please delete the first paragraph in part of 3.1

Response: We have deleted it according to the Reviewer’s comments.

  1. There seems to be some mistakes in the format of the references, which need to be further proofread and improved.

Response: We have revised the format of the references according to the Reviewer’s comments.

  1. What does the height of the shear head in Figure 8 mean.

Response: The height of the arrow represents the change of the indicator. The higher the arrow is, the greater the change is. We had added it in the Figure 8.

Reviewer 2 Report

The manuscript titled Roles of ghrelin and leptin on body mass regulation under food restriction based on AMPK pathway in Eothenomys miletus from Kunming and Dali regions by Liu and co-workers evaluated 36 voles from two regions in China (I think). The  simple summary needs to be simplified. I think adding the common name of Eothenomys miletus would be helpful in the title as well as the simple summary. The regions are not meaningful to a casual reader—describe the regions and add what sets these two populations apart.

This is an associative study where animals were placed on a severe restricted diet. This resulted in fat mass which would cause a decrease in leptin. Ghrelin most likely is increased due to the lack of food in the stomach. Without measures of %Fat Mass and %Fat Free Mass you cannot conclude that leptin and ghrelin are inter-regulatory. These are likely changing independently due to the experimental design.

Other considerations

Line 34. Add experimental n.

Line 107-117. Descriptions of the regions should be included in the methods section

Line 125 – 128. Delete

Line 135. It is not apparent that animals were excluded since 36 were captured and 36 were in the study. It seems likely more animals were captured than are reported in line 129.

Line 166. Perhaps “The collected blood” rather than “The blood taken out”

Line 169. Methods for how stomach ghrelin content was extracted and assayed must be added. This paragraph is only describing plasma samples.

Line 184. Primer details must be added.

Line 223-225. Delete

Line 247 – 248. It is not clear what “relevant indicators of E. miletus” refers to rephrase for clarity.

Table 1 is big and difficult to follow. Suggest dividing into smaller tables for the casual reader to follow.

Line 254. Add “brain” or “hypothalamus” rather than E. miletus

Line 262. mRNA was measured and not protein content. Suggest NPY mRNA expression.

Indicators of significance should be added to the graphs.

Line 278. In the serum protein was measured so “expression” is inappropriate. “serum concentration of leptin” differed.

Line 283 – 289 and throughout . Since these measures are in the serum it should be “serum concentrations” rather than “content”

Line 308. CPT1 (Figure 4c) should be added following malonyl CoA.

Line 359 – 360. This is not clear. This statement should be supported by published research.

Line 365-366. Since ghrelin is synthesized by cells of the stomach, it is not unexpected that ghrelin would be higher in the stomach. This should be expected.

Line 371, 419. Ghrelin does not influence satiety. Ghrelin is a hunger signal and absence of hunger is not necessarily satiety. Leptin plays a role in food intake—not meal to meal, but over the long term, but still does not have a role in satiety. This should be revised.

Line 379. ACC was not measured in the current study so it can not be established that ACC phosphorylation was inhibited by ghrelin. Perhaps you could state this was expected.

Line 384. Food restriction decreased fat mass which decreased leptin. Food restriction does not directly influence leptin. Revise for clarity.

Line 388 – 389. Revise for clarity. What is meant by “…AgRP neurons lead to hanger neurons unable to fullu complete…”

Line 398. This is not established in the present study. Ghrelin and leptin were just measured –regulation of one or the other is not established by the current study design. Revise for clarity.

Author Response

Authors carried out a trial on E. miletus captured in two different regions and investigated the effect of feed restriction and geographical region on marker molecules regarding feed intake regulation.

Specific comments concerning the manuscript:

-Please read your manuscript carefully as eg. the method section still contains sentences: "The Materials and Methods should be described with sufficient details to allow 126 others to replicate and build on the published results. Please note that the publication of 127 your manuscript implicates that you must make all materials, data, computer code, and 128 pro".

Response: We have deleted it according to the Reviewer’s comments.

- Please check 2.6. Body composition determination because it seems you have written the same two times: dry the organs and weight the mass.

Response: We have checked it according to the Reviewer’s comments, the two dry weights are the measurement of body composition and digestive tract organs. We have revised this part in the revision.

- If you captured 36 animals and all of them (6x6) were included in the study, what do you mean "All pregnant, lactating or young individuals were excluded in the present study"? or 0 animal was excluded?

Response: The pregnant, lactating or young individuals captured in the wild were not used in the present study, and 36 animals were all in non pregnant, lactating or young individuals.

- Please explain when did you collect samples or measured parameters? On the same day for all groups? or control group and feed restricted group on the same day and refer group 30 days later?

Response: Body mass, food intake and RMR in E. miletus were measured on day 0, 30 and 60 for the corresponding group (day 0: control group; day 30: 80% food restriction group; day 60: refeeding group), respectively. We had added it in the revision.

- What was the sex of the animals? and the distribution in the groups?

Response: We had added it in the revision, 36 E. miletus (♂18: ♀18), animals were randomly divided into 6 groups: control group between KM and DL (C-KM (♂3: ♀3), C-DL (♂3: ♀3)), 80% food restriction 30 d group (F80%-KM (♂3: ♀3), F80%-DL (♂3: ♀3)) and refeeding 30 d group (Re30d-KM (♂3: ♀3), Re30d-DL (♂3: ♀3)).

- Please indicate the significant differences in the figures, not only in the text.

Response: We have added the significant differences in the figures according to the Reviewer’s comments.

- Fig 2. I think it is not expression but gene expression. Also modify the text.

Response: We have made the revision according to the Reviewer’s comments.

- Check your raw data on gene expression as the differences between expression intensities are very low and you found them to be significant.

Response: We have checked the raw data according to the Reviewer’s comments.

- What kind of animals were used in the correlation analyses? ad libitum fed, feed restricted? refed? (fig 5c, Fig 6c)

Response: All data of animals in the same area, including control group, food restriction group and re feeding group were combined and analyzed.

- Could you provide Cq values for the reference and target genes?

Response: We have added the Cq values for the reference and target genes in the raw data according to the Reviewer’s comments.

Reviewer 3 Report

Authors carried out a trial on E. miletus captured in two different regions and investigated the effect of feed restriction and geographical region on marker molecules regarding feed intake regulation.

Specific comments concerning the manuscript:

-Please read your manuscript carefully as eg. the method section still contains sentences: "The Materials and Methods should be described with sufficient details to allow 126 others to replicate and build on the published results. Please note that the publication of 127 your manuscript implicates that you must make all materials, data, computer code, and 128 pro".

- Please check 2.6. Body composition determination because it seems you have written the same two times: dry the organs and weight the mass.

- If you captured 36 animals and all of them (6x6) were included in the study, what do you mean "All pregnant, lactating or young individuals were excluded in the present study"? or 0 animal was excluded?

- Please explain when did you collect samples or measured parameters? On the same day for all groups? or control group and feed restricted group on the same day and refer group 30 days later?

- What was the sex of the animals? and the distribution in the groups?

- Please indicate the significant differences in the figures, not only in the text.

- Fig 2. I think it is not expression but gene expression. Also modify the text.

- Check your raw data on gene expression as the differences between expression intensities are very low and you found them to be significant.

- What kind of animals were used in the correlation analyses? ad libitum fed, feed restricted? refed? (fig 5c, Fig 6c)

- Could you provide Cq values for the reference and target genes?

Author Response

The manuscript titled Roles of ghrelin and leptin on body mass regulation under food restriction based on AMPK pathway in Eothenomys miletus from Kunming and Dali regions by Liu and co-workers evaluated 36 voles from two regions in China (I think). The simple summary needs to be simplified.

Response: We have simplified the summary according to the Reviewer’s comments.

I think adding the common name of Eothenomys miletus would be helpful in the title as well as the simple summary.

Response: We have added it according to the Reviewer’s comments.

The regions are not meaningful to a casual reader—describe the regions and add what sets these two populations apart.

Response: We have added Table 1 about annual climate patterns in Kunming and Dali according to the Reviewer’s comments.

This is an associative study where animals were placed on a severe restricted diet. This resulted in fat mass which would cause a decrease in leptin. Ghrelin most likely is increased due to the lack of food in the stomach. Without measures of %Fat Mass and %Fat Free Mass you cannot conclude that leptin and ghrelin are inter-regulatory. These are likely changing independently due to the experimental design.

 Other considerations

Line 34. Add experimental n.

Response: We have added it according to the Reviewer’s comments.

Line 107-117. Descriptions of the regions should be included in the methods section

Response: We have added Table 1 about annual climate patterns in Kunming and Dali according to the Reviewer’s comments.

Line 125 – 128. Delete

Response: We have deleted it according to the Reviewer’s comments.

Line 135. It is not apparent that animals were excluded since 36 were captured and 36 were in the study. It seems likely more animals were captured than are reported in line 129.

Response: We have revised it according to the Reviewer’s comments, 36 E. miletus were captured and selected in KM and DL by mouse cagein winter of 2021. 36 voles used in the present study were all in non pregnant, lactating or young individuals.

Line 166. Perhaps “The collected blood” rather than “The blood taken out”

Response: We have made the revision according to the Reviewer’s comments.

Line 169. Methods for how stomach ghrelin content was extracted and assayed must be added. This paragraph is only describing plasma samples.

Response: We have added the methods for how stomach ghrelin content was extracted and assayed according to the Reviewer’s comments.

Line 184. Primer details must be added.

Response: We have added Table 2 about primer details according to the Reviewer’s comments.

Line 223-225. Delete

Response: We have made the revision according to the Reviewer’s comments.

Line 247 – 248. It is not clear what “relevant indicators of E. miletus” refers to rephrase for clarity.

Response: We have made the revision according to the Reviewer’s comments. Relevant indicators of E. miletus refers to wet and dry mass of liver

Table 1 is big and difficult to follow. Suggest dividing into smaller tables for the casual reader to follow.

Response: We have made the revision according to the Reviewer’s comments.

Line 254. Add “brain” or “hypothalamus” rather than E. miletus

Response: We have made the revision according to the Reviewer’s comments.

Line 262. mRNA was measured and not protein content. Suggest NPY mRNA expression.

Response: We have made the revision according to the Reviewer’s comments.

Indicators of significance should be added to the graphs.

Response: We have added the significant differences in the figures according to the Reviewer’s comments.

Line 278. In the serum protein was measured so “expression” is inappropriate. “serum concentration of leptin” differed.

Response: We have made the revision according to the Reviewer’s comments.

Line 283 – 289 and throughout. Since these measures are in the serum it should be “serum concentrations” rather than “content”

Response: We have spited table into 3,4,5 according to the Reviewer’s comments.

Line 308. CPT1 (Figure 4c) should be added following malonyl CoA.

Response: We have made the revision according to the Reviewer’s comments.

Line 359 – 360. This is not clear. This statement should be supported by published research.

Response: We have added the reference according to the Reviewer’s comments. Food restriction decreases BMR, body and organ mass, and cellular energetics, in the Chinese Bulbul (Pycnonotus sinensis).

Line 365-366. Since ghrelin is synthesized by cells of the stomach, it is not unexpected that ghrelin would be higher in the stomach. This should be expected.

Response: We have made the revision according to the Reviewer’s comments.

Line 371, 419. Ghrelin does not influence satiety. Ghrelin is a hunger signal and absence of hunger is not necessarily satiety. Leptin plays a role in food intake—not meal to meal, but over the long term, but still does not have a role in satiety. This should be revised.

Response: We have made the revision according to the Reviewer’s comments. After reviewing this part of the literature, it is confirmed that ghrelin and leptin regulate food intake rather than satiety, which has been revised in the text.

Line 379. ACC was not measured in the current study so it can not be established that ACC phosphorylation was inhibited by ghrelin. Perhaps you could state this was expected.

Response: We have made the revision according to the Reviewer’s comments.

Line 384. Food restriction decreased fat mass which decreased leptin. Food restriction does not directly influence leptin. Revise for clarity.

Response: We have made the revision according to the Reviewer’s comments.

Line 388 – 389. Revise for clarity. What is meant by “…AgRP neurons lead to hanger neurons unable to fullu complete…”

Response: We have made the revision according to the Reviewer’s comments. AMPK is essential for energy homeostasis regulation and glucose sensing by POMC and AgRP neurons, if AMPK activity is reduced, AgRP neurons unable to fully promote eating and cause body mass loss.

Line 398. This is not established in the present study. Ghrelin and leptin were just measured –regulation of one or the other is not established by the current study design. Revise for clarity.

Response: We have deleted the title 4.3 and included it in Part 4.2, which is supposed to provide basic materials for subsequent research.

Round 2

Reviewer 2 Report

The revised manuscript is improved. I have two concerns/changes that must be addressed.

To reflect the current data--RMR did not differ in DL voles with food restriction and/or refeeding. Line 243- 245 should be revised to reflect that result.

The revised manuscript (line 461 - 462) suggests the hypothalamus is "controlling" the release of ghrelin and leptin. I am not aware of this control and if it has been shown this must be cited. It would be expected ghrelin decreased with stomach fill and leptin increased with increased fat mass. Revise.

Author Response

The revised manuscript is improved. I have two concerns/changes that must be addressed.

To reflect the current data--RMR did not differ in DL voles with food restriction and/or refeeding. Line 243- 245 should be revised to reflect that result.

Response: We have added the result according to the Reviewer’s comments: Although food restriction reduced RMR of DL, the effect of food restriction on RMR of three groups in DL was not significant.

The revised manuscript (line 461 - 462) suggests the hypothalamus is "controlling" the release of ghrelin and leptin. I am not aware of this control and if it has been shown this must be cited. It would be expected ghrelin decreased with stomach fill and leptin increased with increased fat mass. Revise.

Response: Thank you for your valuable suggestions, we had revised it, confirmed by literature review, ghrelin concentration in stomach after food restriction was higher than that in serum, the possible reasons was that if threatened by hunger, ghrelin was secreted in large quantities from the stomach fundus. Therefore, it can detected more ghrelin in stomach after food restriction. After refeeding, ghrelin decreased with stomach fill and leptin increased with increased fat mass.

Reviewer 3 Report

Based on the revised manuscript I suggest acceptance in its present form.

Author Response

Based on the revised manuscript I suggest acceptance in its present form.

Response: Special thanks to you for your good comments.